# Genetic Variations and Health-Related Quality of Life (HRQOL): A Genome-Wide Study Approach

**DOI:** 10.3390/cancers13040716

**Published:** 2021-02-10

**Authors:** Araba A. Adjei, Camden L. Lopez, Daniel J. Schaid, Jeff A. Sloan, Jennifer G. Le-Rademacher, Charles L. Loprinzi, Aaron D. Norman, Janet E. Olson, Fergus J. Couch, Andreas S. Beutler, Celine M. Vachon, Kathryn J. Ruddy

**Affiliations:** 1Department of Oncology, Mayo Clinic, Rochester, MN 55905, USA; Adjei.Araba@mayo.edu (A.A.A.); cloprinzi@mayo.edu (C.L.L.); Andreas.s.beutler.md@gmail.com (A.S.B.); 2Department of Health Sciences Research, Mayo Clinic, Rochester, MN 55905, USA; Lopez.Camden@mayo.edu (C.L.L.); schaid@mayo.edu (D.J.S.); jsloan@mayo.edu (J.A.S.); Le-Rademacher.Jennifer@mayo.edu (J.G.L.-R.); Norman.Aaron@mayo.edu (A.D.N.); olsonj@mayo.edu (J.E.O.); Vachon.Celine@mayo.edu (C.M.V.); 3Department of Laboratory Medicine and Pathology, Rochester, MN 55905, USA; Couch.Fergus@mayo.edu

**Keywords:** health-related quality of life, genome-wide association study, global physical and mental health

## Abstract

**Simple Summary:**

Health-related quality of life (HRQOL) is associated with cancer prognosis as well as with age, sex, race, and lifestyle factors, including diet and physical activity. To investigate the hypothesis that HRQOL has genetic underpinnings in patients with cancer, we performed a genome-wide association study to evaluate genetic variants (single nucleotide polymorphisms, SNPs) associated with mental and physical QOL as measured by the PROMIS assessment tool in breast cancer survivors participating in a longitudinal cohort study, the Mayo Clinic Breast Disease Registry (MCBDR). Age and financial concerns were associated with worse physical and mental health, and previous receipt of chemotherapy was associated with worse mental health. SNPs in *SCN10A, LMX1B, SGCD, PARP12,* and *SEMA5A* were associated with physical and mental QOL, but none at the genome-wide significance thresholds of *p* < 5 × 10^−8^.

**Abstract:**

Health-related quality of life (HRQOL) is an important prognostic patient-reported outcome in oncology. Because prior studies suggest that HRQOL is, in part, heritable, we performed a GWAS to elucidate genetic factors associated with HRQOL in breast cancer survivors. Physical and mental HRQOL were measured via paper surveys that included the PROMIS-10 physical and mental health domain scales in 1442 breast cancer survivors participating in the Mayo Clinic Breast Disease Registry (MCBDR). In multivariable regression analyses, age and financial concerns were significantly associated with global physical health (age: *p* = 1.6 × 10^−23^; financial concerns: *p* = 4.8 × 10^−40^) and mental health (age: *p* = 3.5 × 10^−7^; financial concerns: *p* = 2.0 × 10^−69)^. Chemotherapy was associated with worse global mental health (*p* = 0.01). In the GWAS, none of the SNPs reached the genome-wide association significance threshold of 5 × 10^−8^ for associations with either global physical or global mental health, however, a cluster of SNPs in *SCN10A*, particularly rs112718371, appeared to be linked to worse global physical health (*p* = 5.21 × 10^−8^). Additionally, SNPs in *LMX1B, SGCD, PARP12* and *SEMA5A* were also moderately associated with worse physical and mental health (*p* < 10^−6^). These biologically plausible candidate SNPs warrant further study as possible predictors of HRQOL.

## 1. Introduction

Health-related quality of life (HRQOL) is a multifactorial concept of a person’s self-perception of physical, psychological, and social functioning, often subdivided into several domains, including physical health, emotional health, cognitive functioning, fatigue, and pain [1,2] HRQOL correlates with survival in patients with cancer, and is influenced by demographic characteristics, such as age, sex, and race. Lifestyle factors, including diet and physical activity, are also associated with HRQOL [3,4,5,6]. Quality of life in cancer patients is dynamic and is influenced not only by symptoms of the disease and adverse effects of the treatment but by psychological, social, and spiritual factors as well [7]. Additionally, cancer-related financial hardship has been associated with poor HRQOL [8,9].

Multiple biological pathways play a role in HRQOL [2], including via an impact of the dopaminergic pathway on emotional functioning and via an impact of the inflammation pathway on multiple domains of HRQOL. Specifically, links between the inflammatory pathway and both fatigue and pain are well-established [2,3,4,5,6,10].

Genetic variation may underlie individual differences in HRQOL, which are substantial even for patients receiving identical medical care for very similar health problems. There have been reports regarding the relationship between genetic polymorphisms and pain [11,12,13,14], and family and twin studies have identified 30–50% heritability for subjective well-being, depression, and anxiety, and 22–42% heritability for HRQOL [15,16,17]. Genetic underpinnings of inflammatory pathways may be important, with a number of single nucleotide polymorphisms (SNPs) in cytokine genes and the glutathione metabolic pathway found to be associated with QOL in different populations [2,18,19,20]. Although the exact mechanisms by which genetic variants in inflammatory genes contribute to individual variation in QOL are yet to be elucidated, inflammation may increase the severity of symptoms experienced in diseases, such as cancer, impacting functional status and QOL [2,21]. Genetic variants in inflammatory, dopaminergic, serotonergic, and neurotrophic signaling, as well as neuroactive ligand-receptor interaction pathways, have been associated with overall HRQOL, and oxytocin-related genes and genes involved in the serotonergic and dopaminergic pathways appear to play a role in social functioning [1,2,15,18,19,20].

Candidate gene studies in lung cancer have reported that some SNPs in interleukin genes *IL1β*, *IL1RN*, and *IL10* are associated with fatigue (a common driver of impaired HRQOL in patients with cancer) [2,18]. Other SNPs in inflammation pathway genes, such as *PTGS2* and *LTA,* have also been associated with pain and QOL [22,23,24,25]. In addition, SNPs rs3858300, rs10741191, and rs3852507 in *MGMT* (a DNA repair pathway gene) have been associated with overall QOL in patients with lung cancer, and SNPs rs2756109 in *ABCC2* and rs9524885 in *ABCC4* (glutathione metabolic pathway) have been associated with pain [20]. An SNP in p38 MAPK signaling pathway gene, *MEF2B* rs2040562, has also been reported as a genetic determinant of HRQOL in patients with lung cancer [26]. Building on the findings of these candidate gene studies, a genome-wide study in 5142 Swedish women [17] did not find any SNPs significantly associated with QOL (Bonferroni-corrected significance threshold 2.86 × 10^−7^) but observed SNPs with moderate association, including rs17599095 in *FSTL5* with social functioning and rs813299 in *TRPM1* with global health/QOL. In this current study, we aimed to perform a comprehensive GWAS to explore the relationship between genetic variations and HRQOL domains in a large cohort of breast cancer survivors.

## 2. Results

### 2.1. Patient Characteristics

The characteristics of the participants in this HRQOL study are shown in Table 1. At the time of this study, 8317 patients had consented to the Mayo Clinic Breast Disease Registry (MCBDR), as depicted in Figure 1. After exclusion of subjects without available genotyping data (*n* = 4901), those who did not return a follow-up questionnaire (*n* = 1129), and those who had prior cancer diagnosis (*n* = 352), a DCIS diagnosis (*n* = 329) or a metastatic disease diagnosis (*n* = 38), a total of 1568 patients remained. Nine had not responded to questions inquiring about financial concerns, and 117 had not completed the PROMIS-10 instrument, therefore, 126 additional patients were excluded from our analysis.

A final number of 1442 white patients, three of whom were men, were included in this HRQOL GWAS analysis. Their mean age was 53.4 years (standard deviation 11.5) at the time of cancer diagnosis and 61.6 (12.8) years at the time of QOL assessment.

Approximately 54% of the patients had undergone mastectomy, and 69% had received endocrine therapy. Nearly half had undergone radiation (47.8%), and 42% had received chemotherapy, as shown in Table 1.

In multivariable linear regression models, age was significantly associated with both global physical (*p* = 1.6 × 10^−23^) and global mental (*p* = 3.5 × 10^−7^) health (Table 2). A non-linear relationship between age and the PROMIS-10 measures was apparent, hence, a quadratic term for age was included in the models. The model estimates for the quadratic relationship were such that increasing age was associated with better global physical health until age 46, after which, increasing age was associated with worse physical health. Compared to mean global physical health at age 46, the means at ages 30, 50, 70, and 90 were estimated to be lower by 2.7, 0.1, 5.7, and 19.5, respectively, adjusting for other covariates. A similar observation was seen for global mental health except that the peak for mental health was at age 57 instead of 46. Compared to mean global mental health at age 57, the means at ages 30, 50, 70, and 90 were estimated to be lower by 8.5, 0.6, 1.9, and 12.3 points, respectively. Greater financial concern was significantly associated with worse global physical (*p* = 4.8 × 10^−40^) and mental (*p* = 2.0 × 10^−69^) health, with the mean global physical (mental) health worse by 5.8 (7.7), 13.7 (20.9), and 16.7 (31.1) points for those with financial concerns of 2, 6, and 10, respectively, compared to those with financial concerns of 0 (where 0 = no financial concern). Chemotherapy treatment was associated only with worse global mental health (*p* = 0.01), not global physical health.

### 2.2. GWAS Results

Manhattan plots showing the *p*-values for associations of SNPs with PROMIS-10 global physical health and global mental health and adjusting for the covariates described in Table 2 are shown in Figure 2A,B respectively, and the Q-Q plots are shown in Figure 3A,B for global physical and mental health respectively.

The Q-Q plots show that the observed *p*-values fit the null expected values throughout most of the range of *p*-values, suggesting that population stratification was not a concern.

There was no SNP in this study that achieved genome-wide significance (*p* < 5 × 10^−8^) for an association with either global physical or global mental health (Figure 2A,B). The SNPs with *p*-value < 10^−6^ are shown in Table 3, and among them, the one with the strongest association with global physical health was rs112718371 in *SCN10A,* located on chromosome 3p22.2 (*p* = 5.21 × 10^−8^). Nine SNPs on chromosome 14 were also associated with worse global physical health (*p*-values between 2.38 × 10^−7^ and 5.57 × 10^−7^). Eight of them are highly correlated (LD: r^2^ = 0.87–1.0, D’= 1.0) and one (rs7144304) was weakly correlated with the group (LD: r^2^= 0.43–0.49, D’= 0.74–0.85) (Table 3). These nine SNPs lie downstream of *EXOC5*, (Exocyst Complex Component 5), a gene that encodes a multiple protein complex essential for the biogenesis of epithelial cell surface polarity (https://www.ncbi.nlm.nih.gov/ (accessed on 31 December 2020)) and contains an SNP that has been associated with increased alcohol consumption [27].

Global mental health was also associated with several SNPs with *p*-values < 10^−6^ (Table 3). The SNP with the strongest association, rs73813229 (*p* = 2.84 × 10^−7^), is located in *SGCD* on chromosome 5q33.2-q33.3. It is highly correlated with ten other SNPs in *SGCD* (LD: r^2^ = 0.81–1.0, D’ = 0.94–1.0) and, together with four SNPs in *SEMA5A,* also located on chromosome 5p15.31 (LD: r2 and D’ = 1.0), was found to be associated with worse global mental health in the study. *SEMA5A* is downstream of *SGCD,* and the two genes are separated by approximately 146 mega bases.

Three other SNPs associated with worse global mental health were rs71497626, located in *LMX1B* on chromosome 9q33.3, (*p* = 3.17 × 10^−7^), rs1544460, located in *PARP12* on chromosome 7q34 (*p* = 4.28 × 10^−7^), and rs9899933, located in an intergenic region on chromosome 17 (*p*= 7.99 × 10^−7^) (Table 3).

## 3. Discussion

HRQOL, an individual’s perception of his or her well-being (including physical, psychological, social, and spiritual components), depends on the symptoms of a person’s disease and his or her adverse effects of treatment, and also on innate tendencies to experience more or less discomfort and/or dissatisfaction with those symptoms and side effects [7]. Cancer is often associated with poor health outcomes and morbidity, and can negatively impact quality of life even long after treatment finishes [1,17]. Genetic variations may also play a role, predisposing certain patients to worse HRQOL during and after cancer therapy [18,23,28,29,30,31,32].

In this study of mostly white survivors of stage 1–3 breast cancer, age was associated with better physical and mental HRQOL among younger patients (age <46 for physical health, <57 for mental health) and worse HRQOL among older patients. The association of aging with a decline in physical health is well documented, yet other studies evaluating breast cancer survivors have been inconsistent in showing an impact of age on mental and physical HRQOL [33,34,35,36,37]. Greater financial concerns were associated with worse physical and mental HRQOL, similar to other reports [8,9]. Prior receipt of chemotherapy also was associated with worse mental HRQOL despite a median time of 8.3 years (1st quantile, 5.0; 3rd quantile, 10.9) between diagnosis and survey completion. This is interesting because one might expect that the long-term toxicities of chemotherapy (e.g., neuropathy and cardiac dysfunction) might impair physical HRQOL more than mental HRQOL. Therefore, our observation that worse mental HRQOL is associated with chemotherapy toxicity raises the possibility that psychosocial interventions that help reduce the trauma experienced by patients receiving chemotherapy might be needed.

While there was no SNP that was clearly associated with HRQOL at the genome-wide significance *p*-value threshold (5 × 10^−8^), we found some SNPs in genes that encoded proteins with functional importance that were worthy of follow-up (associated with HRQOL at a more lenient threshold of *p* < 10^−6^). SNPs in *SCN10A, LMX1B, SCGD, SEMA5A,* and *PARP12* appear to be possibly associated with worse global physical or mental health using this more lenient threshold. These findings are consistent with prior evidence suggesting that genetic predisposition to diseases relating to immune, neuroendocrine, and cardiovascular systems contribute to depression, well-being, pain, and fatigue [15,21,38,39,40,41,42].

SNP rs112718371 in *SCN10A* was associated with worse global physical HRQOL. *SCN10A* is a sodium voltage-gated channel alpha subunit 10 gene that encodes a tetrodotoxin-resistant channel protein. This protein mediates the voltage-dependent sodium ion permeability of excitable membranes that assumes opened or closed conformations in response to the voltage difference across the membrane (https://www.ncbi.nlm.nih.gov/ (accessed on 31 December 2020)), forming a sodium-selective channel through which sodium ions may pass in accordance with their electrochemical gradient [43]. The SCN10A protein plays a role in neuropathic pain mechanisms [44,45]. An SNP in the sodium-voltage alpha subunit 9 (*SCN9A*) has been previously associated with pain as well [46]. In addition, genetic variations in *SCN10A* have been associated with cardiac conduction abnormalities in hypertrophic cardiomyopathy patients [47,48] and with GI toxicity in taxane-treated patients [49].

We also found four genes with SNPs that showed a strong association with worse global mental health. First, eleven SNPs in *SGCD* were associated with mental HRQOL. This gene encodes a protein that is one of the four known components of the sarcoglycan complex, a subcomplex of the dystrophin-glycoprotein complex (DGC). DGC forms a link between the F-actin cytoskeleton and the extracellular matrix. The SGCD protein is expressed most abundantly in skeletal and cardiac muscle. While this study observed eight *SGCD* SNPs in a tight linkage that were associated with worse global mental health, others have reported that mutations in this gene are associated with autosomal recessive limb-girdle muscular dystrophy [50,51,52] and dilated cardiomyopathy [53]. The limb-girdle muscular dystrophy is an autosomal recessive degenerative myopathy initially affecting the proximal limb-girdle musculature, and muscle from patients show a complete loss of delta-sarcoglycan as well as other components of the sarcoglycan complex [51]. Dilated cardiomyopathy, on the other hand, is a disorder characterized by ventricular dilation and impaired systolic function, resulting in congestive heart failure and arrhythmia, and patients are at risk of premature death [53].

An SNP in *LMX1B* was also strongly associated with worse global mental health. This gene encodes a member of LIM-homeodomain family of proteins and functions as a transcription factor essential for the normal development of dorsal limb structures, the glomerular basement membrane, the anterior segment of the eye, and dopaminergic and serotonergic neurons (https://www.ncbi.nlm.nih.gov/ (accessed on 31 December 2020)). Certain mutations in this gene are associated with the nail-patella syndrome, [54,55] others are involved in the risk of developing nephropathy [56], and some have been implicated in BMI and obesity risk [57,58].

Polymorphisms in *SEMA5A* were associated with worse mental HRQOL in this study. The gene lies on chromosome 5, located approximately 146 mega-bases downstream of *SGCD*. *SEMA5A* belongs to the semaphorin gene family that encodes membrane proteins containing a semaphorin domain and several thrombospondin type-1 repeats. These are involved in axonal guidance during neural development [59]. Specific polymorphisms in *SEMA5A* have been implicated in autism spectrum disorders and Parkinson’s disease [60,61,62]. *SEMA5A* is also implicated in the Cri-du-chat syndrome in which deletion of the short arm of chromosome 5 (5p-) is associated with phenotypic features, including dysmorphic facial features, microcephaly, and intellectual disability [63]. It is possible that patients who carry some of these variants may be less satisfied with their physical appearance due to their facial features, which could play a role in HRQOL. Furthermore, the protein product semaphorin 5A is reported to be up-regulated in glioma, melanoma, pancreatic, breast, and gastric cancer [64], and also is significantly elevated in rheumatoid arthritis [65]. These conditions could account for the role of *SEMA5A* polymorphisms in HRQOL. Additionally, *SEMA5A* could play a role in chemotherapy-induced peripheral neuropathy (CIPN) given that it is important in the guidance of axons in the central and peripheral nervous system and also that semaphorins are involved in diabetic neuropathy [59,66,67].

*PARP12* is the final gene that appeared to be potentially associated with global mental health. This is a Poly (ADP-Ribose) Polymerase Family Member 12 gene that encodes the protein PARP [Poly (ADP-ribose) polymerase], which catalyzes the post-translational modification of proteins by the addition of multiple ADP-ribose moieties [68]. While *PARP12* has been reported as a tumor suppressor that plays an important role in the suppression of hepatocellular carcinoma metastasis [69], it has also been implicated in coronary heart disease and associated with vitiligo [70,71]. In the current study, we observed that the rs1544460 polymorphism was associated with worse global mental health in a cohort of breast cancer survivors.

The magnitudes of the associations found in this study should be considered with the following in mind: Each of the PROMIS-10 scales used in this study is a rescaled average of patient responses to four items, with each response on a scale of 1 to 5. A change by one step (e.g., from 2 = “Fair” to 3 = ”Good”) on any one item corresponds to a 6.25-point change on the 0–100 scale used in this study. Thus, 6.25 points may be considered the minimum clinically relevant change in mean global physical or mental health, 12.5 points (net improvement or worsening by one step on two items, or by two steps on one item) are more clearly of clinical relevance. For several SNPs in this study, the alternate allele was associated with a change in mean physical or mental health of at least 12.5 points. However, the association magnitudes for those SNPs selected based on their small *p*-values will be exaggerated due to winner’s curse bias and should be interpreted cautiously.

There are limitations to this study based on our suboptimal power (small sample size) for a GWAS and because we did not have a validation set with which to verify our findings. Furthermore, we did not have available data on comorbidity for multivariable analyses of contributors to HRQOL. Finally, most of the participants were white, therefore, these findings may not be generalizable to patients of other racial groups.

## 4. Materials and Methods

### 4.1. Patient Cohort

Genotype and follow-up questionnaire data from the Mayo Clinic Breast Disease Registry (MCBDR) provide the analytic basis for this study. The MCBDR is an ongoing clinic-based longitudinal cohort that enrolls patients with breast cancer diagnosed within the year prior and seen at least once at the Mayo Clinic in Rochester, MN. More than 8000 patients had consented to participate in MCBDR between 2003 and 8 July 2020, the date of database freeze for this study, with accrual rates currently approximating 600/year (70–80% of those approached). Consenting participants complete questionnaires at baseline and during follow-up (by mail) and allow intermittent reviews of their medical records and access to tumor tissue when available. For the current study, patients with other prior cancers, DCIS/stage 0 diagnosis, or stage 4/metastatic breast cancer were excluded. All PROMIS-10 data following a cancer recurrence were also excluded. Otherwise eligible patients, who returned at least one pre-recurrence follow-up questionnaire that included the PROMIS-10, and for whom genotyping had been performed, were included in the analyses (Figure 1). All MCBDR participants signed an IRB-approved, informed consent in accordance with federal and institutional guidelines.

### 4.2. Quality of Life (QOL) Assessments

In this study, the Global Physical Health and Global Mental Health summary scores for the validated PROMIS-10 scale were used to assess the HRQOL of patients in the breast registry cohort [72]. The Patient-Reported Outcomes Measurement Information System Global-10 (PROMIS-10) is a 10-question tool that has been calibrated based on census-population norms and measures that evaluate and monitor physical, mental, and social health applicable across chronic illness populations [72,73,74,75,76,77]. We employed an often-used approach that rescaled PROMIS-10 scores, hence, 0 represented minimum (worst) HRQOL and 100 represented maximum (best) HRQOL along the theoretical range of each domain under study [78]. A separate MCBDR survey question developed as a linear analog self-assessment (LASA)-style item [79] asked the respondent to rate his or her financial concerns on a scale of 0 (no concerns) to 10 (constant concerns). For participants with more than one eligible survey, the latest survey results were used in order to prioritize understanding predictors of long-term HRQOL in survivors (rather than the short-term acute effects of surgery, chemotherapy, and/or radiation treatments).

### 4.3. Genotyping, Quality Control, and Imputation

Genomic DNA was extracted from blood samples from the participants of the breast registry study cohort and the DNA samples were genotyped on either of two platforms, the Illumina Infinium OncoArray (https://www.illumina.com (accessed on 31 December 2020)) and the iCOGS chip, a platform specifically designed to evaluate genetic variants associated with the risk of breast, ovarian, and prostate cancer (http://www.cogseu.org/ (accessed on 31 December 2020)) [80,81].

For quality control, SNPs with call rates <95%, Hardy–Weinberg equilibrium *p*-value < 10^−6^, or minor allele frequency (MAF) < 0.01 were excluded. Samples with discrepancies between subject-reported sex and estimated sex from genetic data, or closely related kinship (within first degree relatives) according to KING [82], were excluded. The STRUCTURE software [83] was used to determine population admixture for the patients on the study, together with reference samples (*n* = 585) of known ancestry from the 1000 Genome database that served as population anchors. A single primary ancestry category (African, Asian, or Caucasian) was predicted for each study sample. Principal component analysis was utilized to assess and correct population stratification and unanticipated relatedness. To increase the genome coverage, genotypes (allele dosages) were imputed by the University of Michigan imputation server [84]. SNPs with imputation accuracy r^2^ < 0.3 were excluded. The 7,254,516 imputed SNPs common to the OncoArray and iCOGS-derived datasets (after other data processing steps) were evaluated in analyses.

### 4.4. Statistical Analyses

Linear regression was used to determine potential covariates to be adjusted for in the GWAS, as well as to test the association between each SNP and the trait of interest, adjusted for selected covariates. The covariates considered for selection were age at the time of the PROMIS-10 response, years since cancer diagnosis, self-rated financial concerns, having undergone mastectomy, axillary lymph node dissection (ALND), chemotherapy, radiation therapy, or endocrine therapy, the first five principal components of the genotype data, and genotyping platform (iCOGS versus OncoArray). We selected covariates with *p* < 0.1 in a linear regression of PROMIS-10 (either global physical or global mental health) on covariates only. Quadratic terms for age and financial concerns were included in the regression after observing non-linear relationships in preliminary analysis. Age and financial concerns were transformed to z-scores (by subtracting the sample mean and dividing by the sample standard deviation) to avoid a high correlation between the linear and quadratic terms. We adjusted for genotyping platform in the GWAS despite *p*-values > 0.1 in the covariate regression as a precaution against confounding. SNP genotypes were modeled using the dose of the alternate (minor) allele and assuming an additive effect. Genome-wide significance was defined as *p* < 5 × 10^−8^ [85,86]. Quantile-quantile (Q-Q) plots were used to visually evaluate whether population stratification was controlled by plotting the distribution of observed *p*-values versus the distribution expected under a null hypothesis of no SNP associations. Manhattan plots were used to plot *p*-values for all SNP associations across chromosomes, and regional association plots (Locus Zoom) [87] were used to provide detail on genetic regions of interest, providing gene annotations and pairwise correlations between the surrounding SNPs and the SNP of interest. Additionally, Ldlink [88,89], a suite of web-based applications, was used to efficiently interrogate linkage disequilibrium (LD) for SNPs.

## 5. Conclusions

Although we did not identify that physical or mental HRQOL was associated with any SNP at the genome-wide significant threshold, we did find weaker associations with some biologically plausible SNPs that should be assessed further in future research.

## Figures and Tables

**Figure 1 cancers-13-00716-f001:**
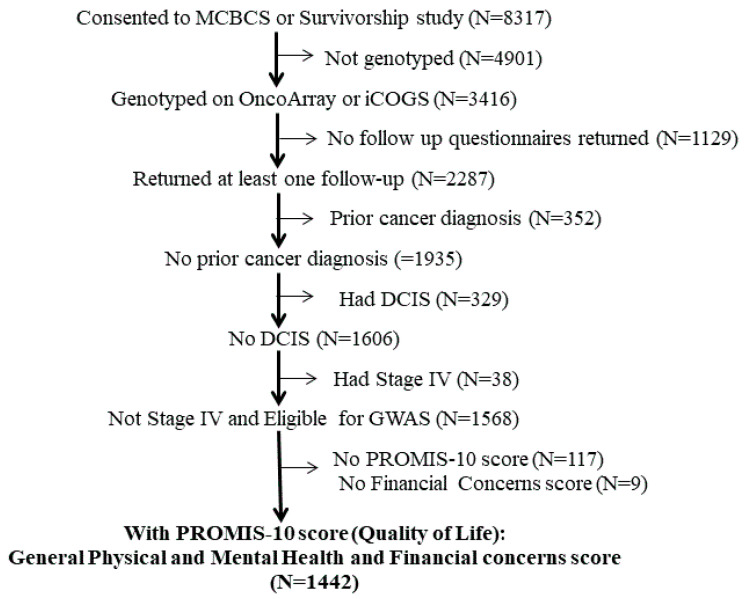
Breast Registry Flow Diagram.

**Figure 2 cancers-13-00716-f002:**
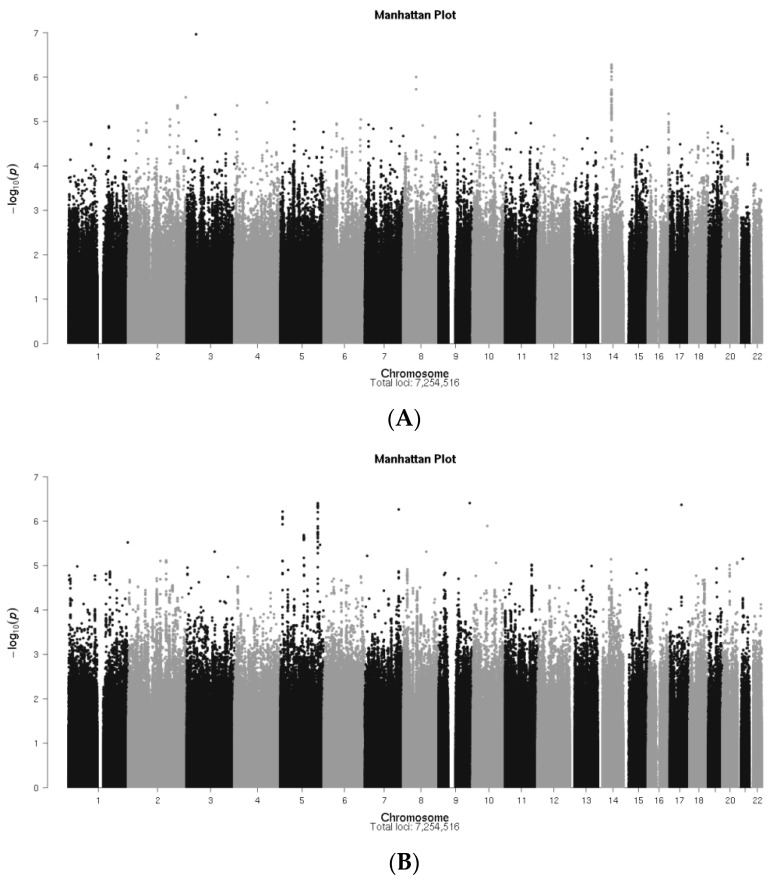
(**a**) Manhattan plot for genome-wide associations with PROMIS-10 global physical health in MCBDR cohort. (**b**) Manhattan plot for genome-wide associations with PROMIS-10 global mental health in MCBDR cohort.

**Figure 3 cancers-13-00716-f003:**
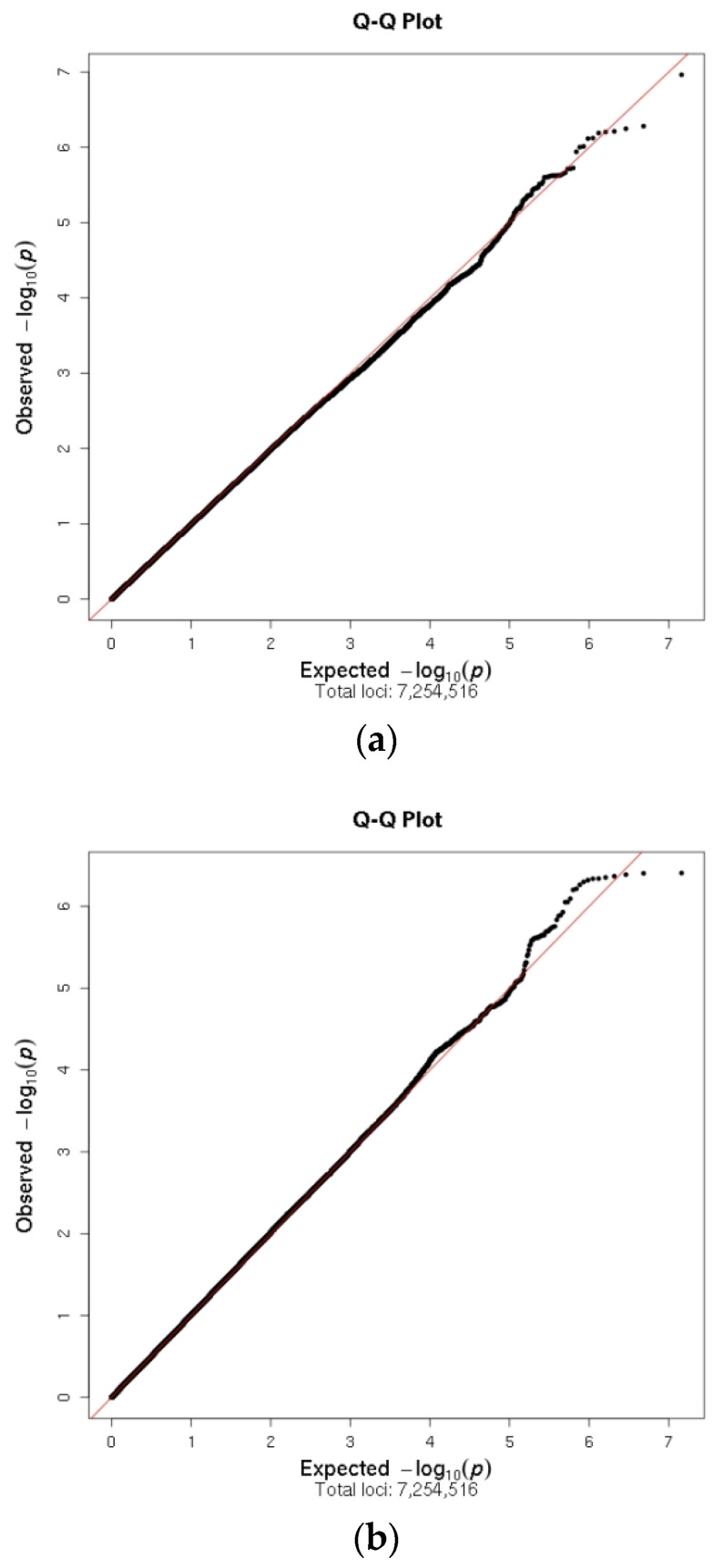
(**A**) Q-Q plot for genome-wide associations with PROMIS-10 global physical health in MCBDR cohort. (**B**) Q-Q plot for genome-wide associations with PROMIS-10 global physical health in MCBDR cohort.

**Table 1 cancers-13-00716-t001:** Patient characteristics (*n* = 1442).

Demographic Factors		*n* (%)
Race	Non-white, unknown, or undisclosed	56 (3.9%)
	White	1386 (96.1%)
Age at diagnosis in years	Mean (SD)	53.4 (11.5)
	Median (Q1, Q3)	51.5 (45.6, 62.4)
	Range	22.7–90.0
Age at time of QOL assessment in years	Mean (SD)	61.6 (12.8)
	Median (Q1, Q3)	61.0 (52.5, 71.4)
	Range	26.2–95.5
Years between cancer diagnosis and QOL assessment	Mean (SD)	8.2 (3.8)
	Median (Q1, Q3)	8.3 (5.0, 10.9)
	Range	0.9–16.7
Gender	Male	3 (0.2%)
	Female	1439 (99.8%)
Financial concerns rating (patient-reported)	Mean (SD)	2.1 (2.7)
	Median (Q1, Q3)	1.0 (0.0, 3.0)
	Range	0.0–10.0
Treatment received any time before QOL assessment (Yes/No)	Mastectomy	782 (54.2%)/660 (45.8%)
	Axillary lymph node dissection (ALND)	530 (36.8%)/912 (63.2%)
	Chemotherapy	608 (42.2%)/834 (57.8%)
	Radiation	689 (47.8%)/753 (52.2%)
	Endocrine therapy	997 (69.1%)/445 (30.9%)

**Table 2 cancers-13-00716-t002:** Multivariable regression analysis: Factors related to HRQOL in the MCBDR.

HRQOL (PROMIS-10 Measure)	Covariate	Coefficient			^b ^*p*-Value
		Estimate	SE	^a^*p*-Value	
Global Physical Health	Genotyped with iCOGS	0.63	0.83	0.45	-
	Age ^c^ z-score	−4.00	0.45	2.0 × 10^−18^	1.6 × 10^−23^
	Age z-score, squared	−1.67	0.32	3.0 × 10^−7^	
	Financial concerns z-score	−6.93	0.64	6.0 × 10^−26^	4.8 × 10^−40^
	Financial concerns z-score, squared	1.12	0.37	0.002	
	Chemotherapy treatment	−0.59	0.83	0.48	-
Global Mental Health	Genotyped with iCOGS	−0.05	0.99	0.96	-
	Age z-score	−1.27	0.54	0.02	3.5 × 10^−7^
	Age z-score, squared	−1.88	0.39	1.0 × 10^−6^	
	Financial concerns z-score	−9.83	0.77	2.0 × 10^−35^	2.0 × 10^−69^
	Financial concerns z-score, squared	0.67	0.44	0.13	
	Chemotherapy treatment	−2.47	1.00	0.01	-

^a^ Regression results are shown after selecting covariates based on *p* < 0.1, ^b^ Overall *p*-value from a Wald test, ^c^ Age and financial concerns were transformed to z-scores (by subtracting the sample mean and dividing by the sample standard deviation) to avoid high correlation between the linear and quadratic terms, SE: standard error.

**Table 3 cancers-13-00716-t003:** SNPs with *p*-values < 10^−6^ for associations with PROMIS-10 global physical and mental health.

QOL Domain	Gene	SNP rsID	Chr	BP	Ref Allele	Alt Allele	Alt Allele Freq	Coefficient (β) Estimate	SE	*p*-Value
Global Physical Health	*SCN10A*	rs112718371	3	38750460	C	T	0.011	−17.79	3.25	5.21 × 10^−8^
	*-*	rs79198292	14	57649431	G	A	0.022	−10.46	2.01	2.38 × 10^−7^
		rs1954548	14	57644518	G	A	0.023	−10.33	1.99	2.62 × 10^−7^
		rs1954547	14	57644605	C	T	0.023	−10.27	1.99	2.91 × 10^−7^
		rs2211582	14	57643873	C	T	0.022	−10.40	2.02	2.97 × 10^−7^
		rs73300594	14	57641616	C	T	0.023	−10.31	2.01	3.50 × 10^−7^
		rs60372931	14	57642708	A	G	0.023	−10.29	2.01	3.56 × 10^−7^
		rs7159115	14	57645169	A	G	0.023	−10.09	1.99	4.58 × 10^−7^
		rs11848373	14	57649554	G	A	0.024	−9.86	1.96	5.51 × 10^−7^
		rs7144304	14	57655035	A	T	0.034	−8.08	1.61	5.57 × 10^−7^
Global Mental Health	*SGCD*	rs73813229	5	155504517	T	C	0.185	−5.10	0.99	2.84 × 10^−7^
		rs73298688	5	155508728	C	T	0.184	−5.10	0.99	2.99 × 10^−7^
	*LMX1B*	rs71497626	9	129405863	G	A	0.011	−17.16	3.34	3.17 × 10^−7^
	*SGCD*	rs113472609	5	155501582	T	C	0.186	−5.07	0.99	3.17 × 10^−7^
		rs80138336	5	155501438	A	G	0.185	−5.06	0.99	3.27 × 10^−7^
		rs73813228	5	155501356	A	G	0.185	−5.06	0.99	3.29 × 10^−7^
		rs4704970	5	155500992	G	A	0.185	−5.05	0.99	3.42 × 10^−7^
		rs73813227	5	155500542	T	C	0.185	−5.04	0.99	3.59 × 10^−7^
	*PARP12*	rs1544460	7	139731709	G	A	0.430	−3.41	0.67	4.28 × 10^−7^
	*SGCD*	rs10476276	5	155500647	A	G	0.188	−4.96	0.98	4.50 × 10^−7^
	*SEMA5A*	rs76677754	5	9084176	C	T	0.021	−14.26	2.81	4.58 × 10^−7^
		rs11741186	5	9044674	G	A	0.020	−13.26	2.65	6.13 × 10^−7^
	*SGCD*	rs73812917	5	155464275	G	A	0.186	−4.84	0.97	6.15 × 10^−7^
	*SEMA5A*	rs78456783	5	9037340	A	C	0.020	−13.21	2.65	6.76 × 10^−7^
	*-*	rs9899933	17	48032244	A	G	0.060	−8.35	1.69	7.99 × 10^−7^
	*SGCD*	rs75174473	5	155462439	A	G	0.203	−4.61	0.93	8.84 × 10^−7^
	*SEMA5A*	rs11741172	5	9044647	G	A	0.020	−13.24	2.68	9.02 × 10^−7^
	*SGCD*	rs58327079	5	155463960	T	C	0.203	−4.60	0.93	9.93 × 10^−7^

*LMX1B*: LIM homeobox transcription factor 1 beta, Location: Chromosome 9q33.3; *PARP12*: Poly(ADP-ribose) polymerase family member 12, Location: Chromosome 7q34; *SCN10A*: Sodium voltage-gated channel alpha subunit 10, Location: Chromosome 3p22.2; *SEMA5A*: Semaphorin 5A, Location: Chromosome 5p15.31; *SGCD*: Sarcoglycan delta, Location: Chromosome 5q33.2-q33.3; freq: Frequency; SE: standard error.

## Data Availability

The datasets used and/or analyzed during the current study are available from the corresponding author on reasonable request.

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
