# Peer review of "Genetic Variations and Health-Related Quality of Life (HRQOL): A Genome-Wide Study Approach"

_cancers, 2021, doi:10.3390/cancers13040716_

Round 1

Reviewer 1 Report

In this study, HRQoL outcomes have been associated with genetic variants. This is an interesting study, with a large number of patients with respect to the clinical variables. The results show that several sociodemographic and clinical variables were independently associated with physical or mental health, but that the genetic variation were not associated at the genome-wide significance threshold.

I have several comments:

  • Health-related quality of life (HRQOL) refers to a person’s self-perception of physical, psychological, and social functioning, and is impacted by domains including physical health, emotional health, cognitive functioning, social functioning, fatigue, and pain.’ How can social functioning be impacted by social functioning? It is better to state that HRQoL is a multidimensional concept, coverings aspects of physical, psychological and social functioning, as well as symptoms. Since HRQoL is a multidimensional concept, it is not correct to state that ‘HRQOL is impacted by symptoms’ or that ‘psychological and social status influence HRQOL’, as this are aspects of HRQoL. In that case you are probably referring to the general health status or score for overall quality of life, as symptoms can impact functioning (e.g. social functioning) and overall QoL. Overall, the term HRQoL should be used in the correct manner.
  • What is the definition of a breast cancer survivor? The range in years between diagnosis and HRQoL assessment varies between 0.9 years and 16.7 years. I wonder if you could classify a breast cancer patient 0.9 years after diagnosis as a survivor. In most definitions, patients need to be a certain amount of time after finishing treatment (except maintenance hormonal therapy).
  • What was the rationale for selecting financial concerns as covariate? And the other variables? It would be helpful if it is explained that these were selected based on clinical experience and/or previous studies (in that case with references).
  • This sentence does not seem to be finished? ‘SNP genotypes were modeled using the dose of the minor allele and assuming an additive effect, with.’
  • 1442/8317 (17%) were included in this study. Were these patients different from those that were excluded? This could be checked for a few known factors, such as age, gender, type of treatment. I will demonstrate how generalizable the result are.
  • Were multivariate or multivariable analyses performed? Thus, with multiple dependent variables or independent variables, respectively?
  • Were the linear and quadratic variables included in one model?
  • In the results section, age is suddenly categorized (under 46 or 57). How was this included in the analyses?
  • Currently, the results are only interpreted in terms of statistical significance. It would be more relevant if the results are also interpreted in terms of clinical relevance. For example, for the SNPs associated with physical or mental health, Beta’s vary roughly between 5 and 15 points. What is the minimal clinical important difference for these questionnaires? In other words, are the found differences clinically relevant. Similar for the sociodemographic variables, what do the associations mean? How strong are they?
  • The authors suggest that long-term toxicity of chemotherapy is the reason for the lower mental HRQoL in breast cancer survivors. Are they sure that all confounding factors are included? I can imagine that those patients receiving chemotherapy have a different profile than those not needing chemotherapy. Thus, can this finding be attributed to long-term toxicity or is this a result of bias?
  • In the discussion, the function of each of the SNPs that were weakly associated with physical or mental health were described. Most of these SNPs were previously found to be associated with other diseases (e.g. heart disease or rheumatoid arthritis).
    • How will this translate into worse physical, and particularly mental, health? Can the authors elaborate on that?
    • Was information about comorbidity known for the included patients? I wonder if you would include comorbidity into the multivariable analysis, whether you would still find these associations?
  • What is the clinical value of these results? How to continue?

Reviewer 2 Report

This is very interesting study because it investigates very distant phenomena - highly subjective overall evaluation of person's quality of life and biological microscale variables from the GWAS. Although some relationships between these psychological and genetics realms might be expected, they cannot be strong or straightforward by default, and this should be more clearly acknowledged. Thus, I would suggest several improvements:

  1. Please describe concept of quality of life in Introduction. PROMIS is a very general instrument, not specific to brain cancer symptoms. It asks for overall evaluation of different aspects of persons functioning, and this evaluation is influence by the myriads of factors, including personality, relationships, social context, expectations etc. Please provide arguments, why it is worth investigating genetic underpinnings of of such evaluations. What would be the clinical and scientific value of such findings?
  2. Please provide some clarifications regarding age analysis. Older age is usually associated with worse QoL both in patients and general population. However, in current study it was hard to understand, why certain cutoffs are reported and why they were chosen for the analysis (for example: "In patients under age 46, global physical health was better in older patients".  "In patients under age 57, global mental health was better in older patients"). Please discuss current findings in Discussion section. They are possibly related to the course of illness and age at the onset of the illness. I think that age at diagnosis and stage  are worth including into analysis, because they are predictive to course of the illness. 
  3. When discussing findings QoL relationships with specific SNP, please focus on relations that are meaningful. For example, SEMA5A is related to dismorphic facial features and microcephalus. How does this information help to understand correlation with QoL ratings in breast cancer? Provide more insight about possible mechanisms of genetic underpinnings of QoL in breast cancer. 
